# Streaming Bayesian Inference for Crowdsourced Classification

**Edoardo Manino**
University of Southampton
E.Manino@soton.ac.uk

**Long Tran-Thanh**
University of Southampton
l.tran-thanh@soton.ac.uk

**Nicholas R. Jennings**
Imperial College, London
n.jennings@imperial.ac.uk

## Abstract

A key challenge in crowdsourcing is inferring the ground truth from noisy and unreliable data. To do so, existing approaches rely on collecting redundant information from the crowd, and aggregating it with some probabilistic method. However, oftentimes such methods are computationally inefficient, are restricted to some specific settings, or lack theoretical guarantees. In this paper, we revisit the problem of binary classification from crowdsourced data. Specifically we propose Streaming Bayesian Inference for Crowdsourcing (SBIC), a new algorithm that does not suffer from any of these limitations. First, SBIC has low complexity and can be used in a real-time online setting. Second, SBIC has the same accuracy as the best state-of-the-art algorithms in all settings. Third, SBIC has provable asymptotic guarantees both in the online and offline settings.

## 1 Introduction

Crowdsourcing works by collecting the annotations of large groups of human workers, typically through an online platform like Amazon Mechanical Turk[1] or Figure Eight.[2] On one hand, this paradigm can help process high volumes of small tasks that are currently difficult to automate at an affordable price [Snow *et al.*, 2008]. On the other hand, the open nature of the crowdsourcing process gives no guarantees on the quality of the data we collect. Leaving aside malicious attempts at thwarting the result of the crowdsourcing process [Downs *et al.*, 2010], even well-intentioned crowdworkers can report incorrect answers [Ipeirotis *et al.*, 2010].

Thus, the success of a crowdsourcing project relies on our ability to reconstruct the ground-truth from the noisy data we collect. This challenge has attracted the attention of the research community which has explored a number of algorithmic solutions. Some authors focus on probabilistic inference on graphical methods, including the early work of Dawid and Skene [1979] on EM estimation, variational inference [Welinder and Perona, 2010; Liu *et al.*, 2012] and belief propagation [Karger *et al.*, 2014]. These techniques are stable in most settings, easy to generalise to more complex models (e.g. [Kim and Ghahramani, 2012]), but generally require several passes over the entire dataset to converge and lack theoretical guarantees. In contrast, other authors have turned to tensor factorisation [Dalvi *et al.*, 2013; Zhang *et al.*, 2016] and the method of moments [Bonald and Combes, 2017]. This choice yields algorithms with tractable theoretical behaviour, but the assumptions required to do so restrict them to a limited number of settings.

At the same time, there have been several calls to focus on how we sample the data from the crowd, rather than how we aggregate it [Welinder and Perona, 2010; Barowy *et al.*, 2012; Simpson and Roberts, 2014; Manino *et al.*, 2018]. All of these end up recommending some form of adaptive strategy, which samples more data on the tasks where the crowd is disagreeing the most. Employing one of these strategies improves the final accuracy of the crowdsourcing system, but requires the ability to work in an online setting. Thus, in order to perform crowdsourcing effectively, our algorithms must be computationally efficient.

In this paper, we address these research challenges on the problem of binary classification from crowdsourced data, and make the following contributions to the state of the art.:

- We introduce Streaming Bayesian Inference for Crowdsourcing (SBIC), a new algorithm based on approximate variational Bayes. This algorithm comes in two variants.

- The first, Fast SBIC, has similar computational complexity to the quick majority rule, but delivers more than an order of magnitude higher predictive accuracy.

- The second, Sorted SBIC, is more computationally intensive, but delivers state-of-the-art predictive accuracy in all settings.

- We quantify the asymptotic performance of SBIC in both the offline and online setting analytically. Our theoretical bounds closely match the empirical performance of SBIC.

The paper is structured in the following way. In Section 2 we introduce the most popular model of crowdsourced classification, and the existing aggregation methods. In Section 3 we present the SBIC algorithm in its two variants. In Section 4 we compute its asymptotical accuracy. In Section 5 we compare its performance with the state of the art on both synthetic and real-world datasets. In Section 6 we conclude and outline possible future work.

## 2 Preliminaries

Existing works in crowdsourced classification are mostly built around the celebrated Dawid-Skene model [Dawid and Skene, 1979]. In this paper we adopt its binary, or one-coin variant, which has received considerable attention from the crowdsourcing community [Liu *et al.*, 2012; Karger *et al.*, 2014; Bonald and Combes, 2017; Manino *et al.*, 2018]. The reason for this is that it allows to study the fundamental properties of the crowdsourcing process, without dealing with the peculiarities of more complex scenarios. Furthermore, generalising to the multi-class case is usually straightforward (e.g. [Gao *et al.*, 2016]).

### 2.1 The one-coin Dawid-Skene model

According to this model, the objective is to infer the binary ground-truth class $y_i = \{\pm 1\}$ of a set tasks $M$, with $i \in M$. To do so, we can interact with the crowd of workers $N$, and ask them to submit a set of labels $X = \{x_{ij}\}$, where $j \in N$ is the worker's index. We have no control on the availability of the workers, and we assume that we interact with them in sequential fashion. Thus, at each time step $t$ a single worker $j = a(t)$ becomes available, gets assigned to a task $i$ and provides the label $x_{ij} = \pm 1$ in exchange for a unitary payment. We assume that we can collect an average of $R \leq |N|$ labels per task, for a total budget of $T = R|M|$ labels. With slight abuse of notation, we set $x_{ij} = 0$ for any missing task-worker pairs, so that we can treat $X$ as a matrix when needed. On a similar note, we use $M_j$ to denote the set of tasks executed by worker $j$, and $N_i$ for the set of workers on task $i$. Furthermore, we use the superscript $t$ (e.g. $X^t$) to denote the information visible up to time $t$.

A key feature of the one-coin Dawid-Skene model is that each worker has a fixed probability $\mathbb{P}(x_{ij} = y_i) = p_j$ of submitting a correct label. That means that the workers behave like independent random variables (conditioned on the ground-truth $y_i$), and their accuracy $p_j$ remains stable over time and across different tasks.

### 2.2 Sampling the data

When interacting with the crowd, we need to decide which tasks to allocate the incoming workers to. The sampling policy $\pi$ we use to make these allocations has a considerable impact on the final

accuracy of our predictions, as demonstrated by Manino *et al.* [2018]. The existing literature provides us with the following two main options.

**Uniform Sampling (UNI).**   This policy allocates the same number of workers $|N_i| \approx T/|M|$ to each task $i$ (rounded to the closest integer). The existing literature does not usually specify how this policy is implemented in practice (e.g [Karger *et al.*, 2014; Manino *et al.*, 2018]). In this paper we assume a round-robin implementation, where we ensure that no worker is asked to label the same task twice:

$$\pi_{uni}(t) = \underset{i \notin M_{a(t)}^t}{\operatorname{argmin}} \left\{ |N_i^t| \right\} \tag{1}$$

where $M_{a(t)}^t$ is the set of tasks labelled by the currently available worker $j = a(t)$ so far.

**Uncertainty Sampling (US).**   A number of policies proposed in the literature are *adaptive*, in that they base their decisions on the data collected up to time $t$ [Welinder *et al.*, 2010; Barowy *et al.*, 2012; Simpson and Roberts, 2014]. In this paper we focus on the most common of them, which consist of greedily choosing the task with the largest uncertainty at each time-step $t$. More formally, assume that we have a way to estimate the posterior probability on the ground-truth $\boldsymbol{y}$ given the current data $X^t$. Then, we can select the task to label as follows:

$$\pi_{us}(t) = \underset{i \notin M_{a(t)}^t}{\operatorname{argmin}} \left\{ \max_{\ell \in \{\pm 1\}} \left( \mathbb{P}(y_i = \ell | X^t) \right) \right\} \tag{2}$$

Compared to uniform sampling, this second policy is provably better [Manino *et al.*, 2018]. However, it can only be implemented in an online setting, when we have estimates of the posterior on $\boldsymbol{y}$ at every $t$. Producing such estimates in real time is an open challenge. Current approaches are based on simple heuristics like the majority voting rule [Barowy *et al.*, 2012].

We study the theoretical and empirical performance of SBIC under these two policies in Sections 4 and 5 respectively.

## 2.3   Aggregating the data

Given a (partial) dataset $X^t$ as input, there exist several methods in the literature to form a prediction $\hat{\boldsymbol{y}}$ over the ground-truth classes $\boldsymbol{y}$ of the tasks. The simplest is the aforementioned majority voting rule (MAJ), which forms its predictions as $\hat{y}_i = \operatorname{sign}\{\sum_{j \in N_i} x_{ij}\}$, where ties are broken at random.

Alternatively, we can resort to Bayesian methods, which infer the value of the latent variables $\boldsymbol{y}$ and $\boldsymbol{p}$ by estimating their posterior probability $\mathbb{P}(\boldsymbol{y}, \boldsymbol{p}|X, \theta)$ given the observed data $X$ and prior $\theta$. In this regard, Liu *et al.* [2012] propose an approximate variational mean-field algorithm (AMF) and show its similarity to the original expectation-maximisation (EM) algorithm of Dawid and Skene [1979]. Conversely, Karger *et al.* [2014] propose a belief-propagation algorithm (KOS) on a spammer-hammer prior, and show its connection to matrix factorisation. Both these algorithms require several iterationd on the whole dataset $X$ to converge to their final predictions. As another option, we can directly estimate the value of the posterior by Montecarlo Sampling (MC) [Kim and Ghahramani, 2012], even though this is usually more expensive computationally than the former two techniques.

Finally, there have been attempts at applying the frequentist approach to crowdsourcing [Dalvi *et al.*, 2013; Zhang *et al.*, 2016; Bonald and Combes, 2017]. The resulting algorithms have tractable theoretical properties, but put strong constraints on the rank and sparsity of the task-worker matrix $X$, which limit their range of applicability. For completeness, we include in our experiments of Section 5 the Triangular Estimation algorithm (TE) recently proposed in [Bonald and Combes, 2017].

## 3   The SBIC algorithm

In this section we introduce Streaming Bayesian Inference for Crowdsourcing (SBIC) and discuss the ideas behind it. Then, we present two variants of this method, which we call Fast SBIC and Sorted SBIC. These prioritise two different goals: namely, computational speed and predictive accuracy.

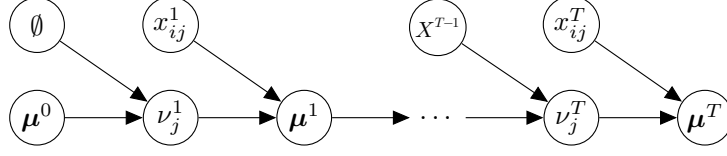

Figure 1: A graphical representation of the SBIC algorithm.

The overarching goal in Bayesian inference is estimating the posterior probability $\mathbb{P}(\boldsymbol{y}, \boldsymbol{p}|X^t, \theta)$ on the latent variables $\boldsymbol{y}$ and $\boldsymbol{p}$ given the data we observed so far $X^t$ and the prior $\theta$. With this piece of information, we can form our current predictions $\hat{\boldsymbol{y}}^t$ on the task classes by looking at the marginal probability over each $y_i$ as follows:

$$\hat{y}_i^t = \underset{\ell \in \{\pm 1\}}{\operatorname{argmax}} \left\{ \mathbb{P}(y_i = \ell | X^t, \theta) \right\} \tag{3}$$

Unfortunately the marginal in Equation 3 is computationally intractable in general. In fact, just summing $\mathbb{P}(\boldsymbol{y}, \boldsymbol{p}|X^t, \theta)$ over all vectors $\boldsymbol{y}$ that contain a specific $y_i$ has exponential time complexity in $|M|$. To overcome this issue, we turn to a mean-field variational approximation as done before in [Liu *et al.*, 2012; Kim and Ghahramani, 2012]. This allows us to factorise the posterior as follows:

$$\mathbb{P}(\boldsymbol{y}, \boldsymbol{p}|X^t, \theta) \approx \prod_{i \in M} \mu_i^t(y_i) \prod_{j \in N} \nu_j^t(p_j) \tag{4}$$

where the factors $\mu_i^t$ correspond to each task $i$ and the factors $\nu_j^t$ to each worker $j$.

Our work diverges from the standard variational mean field paradigm [Murphy, 2012] in that we use a novel method to optimise the factors $\boldsymbol{\mu}^t$ and $\boldsymbol{\nu}^t$. Previous work minimises the Kullback-Leibler (KL) divergence between the two sides of Equation 4 by running an expensive coordinate descent algorithm with multiple passes over the whole dataset $X^t$ [Liu *et al.*, 2012; Kim and Ghahramani, 2012]. Instead, we aim at achieving a similar result by taking a single optimisation step after observing each new data point. This yields quicker updates to $\boldsymbol{\mu}^t$ and $\boldsymbol{\nu}^t$, thus allowing us to run our algorithm online.

More specifically, the core ideas of the SBIC algorithm are the following. First, assume that the prior on the worker accuracy is $p_j \sim \text{Beta}(\alpha, \beta)$. This assumption is standard in Bayesian statistics, since the Beta distribution is the conjugate prior of a Bernoulli-distributed random variable [Murphy, 2012]. Second, initialise the task factors $\boldsymbol{\mu}^0$ to their respective prior $\mathbb{P}(y_i = +1) = q$, that is $\mu_i^0(+1) = q$ and $\mu_i^0(-1) = 1 - q$ for all $i \in M$.[3] Then, upon observing a new label at time $t$, update the factor $\nu_j^t$ corresponding to the current available worker $j = a(t)$ only. Thanks to the properties of the KL divergence, $\nu_j^t$ is still Beta-distributed:

$$\nu_j^t(p_j) \sim \text{Beta}\Big( \sum_{i \in M_j^{t-1}} \mu_i^{t-1}(x_{ij}) + \alpha, \ \sum_{i \in M_j^{t-1}} \mu_i^{t-1}(-x_{ij}) + \beta \Big) \tag{5}$$

where $M_j^{t-1}$ is the set of tasks labelled by worker $j$ up to time $t-1$. Next, we update the factor $\mu_i$ corresponding to the task we observed the new label $x_{ij}$ on:

$$\mu_i^t(y_i) \propto \begin{cases} \mu_i^{t-1}(y_i)\bar{p}_j^t & \text{if } x_{ij} = y_i \\ \mu_i^{t-1}(y_i)\big(1 - \bar{p}_j^t\big) & \text{if } x_{ij} \neq y_i \end{cases} \qquad \text{where} \qquad \bar{p}_j^t = \frac{\sum_{i \in M_j^{t-1}} \mu_i^{t-1}(x_{ij}) + \alpha}{|M_j^{t-1}| + \alpha + \beta} \tag{6}$$

Finally, we can inspect the factors $\boldsymbol{\mu}^t$ and form our predictions on the task classes as $\hat{y}_i^t = \operatorname{argmax}_{\ell \in \{\pm 1\}} \{\mu_i^t(\ell)\}$. Note that we set $\bar{p}_j^t = \mathbb{E}_{p_j}\{\nu_j^t\}$ in Equation 6. An exact optimisation step would require $\bar{p}_j^t = \exp\big(\mathbb{E}_{p_j}\{\log(\nu_j^t)\}\big)$ instead. However, the first-order approximation we use has a negligible impact on the accuracy of the inference, as demonstrated in [Liu *et al.*, 2012].

We summarise the high-level behaviour of the SBIC algorithm in the explanatory sketch of Figure 1. There, it is easy to see that SBIC falls under the umbrella of the Streaming Variational Bayes framework [Broderick *et al.*, 2013]: in fact, at each time step $t$ we trust our current approximations

**Algorithm 1** Fast SBIC

---

**Input**: dataset $X$, availability $a$, policy $\pi$, prior $\theta$
**Output**: final predictions $\hat{\boldsymbol{y}}^T$

1: $z_i^0 = \log(q/(1-q))$, $\forall i \in M$
2: **for** $t = 1$ **to** $T$ **do**
3:     $i \leftarrow \pi(t)$
4:     $j \leftarrow a(t)$

5:     $\bar{p}_j^t \leftarrow \dfrac{\sum_{h \in M_j^{t-1}} \mathrm{sig}(x_{hj} z_h^{t-1}) + \alpha}{|M_j^{t-1}| + \alpha + \beta}$
6:     $z_i^t \leftarrow z_i^{t-1} + x_{ij} \log(\bar{p}_j^t/(1-\bar{p}_j^t))$
7:     $z_{i'}^t \leftarrow z_{i'}^{t-1}$, $\forall i' \neq i$
8: **return** $\hat{y}_i^T = \mathrm{sign}(z_i^T)$, $\forall i \in M$

---

$\boldsymbol{\mu}^t$ and $\boldsymbol{\nu}^t$ to be close to the exact posterior, and we use their values to inform the next local updates. From another point of view, SBIC is a form of constrained variational inference, where the constraints are implicit in the local steps we make in Equations 5 and 6, as opposed to an explicit alteration of the KL objective. Finally, the sequential nature of the SBIC algorithm means that its output is deeply influenced by the order in which we process the dataset in $X$. By altering its ordering, we can optimise SBIC for different applications, as we show in the next two Sections 3.1 and 3.2.

### 3.1 Fast SBIC

Recall that crowdsourcing benefits from an online approach, since it allows the deployment of an adaptive sampling strategy which can greatly improve the predictive accuracy (see Section 2.2). Thus, our main goal here is computational speed, which we achieve by keeping the natural ordering of the set $X$ unaltered.

We call the resulting algorithm Fast SBIC, and show its pseudocode in Algorithm 1. There, we use the following computational tricks. First, we express the value of each factor $\mu_i^t$ in terms of its log-odds. Accordingly, Equation 6 becomes:

$$z_i^t = \log\left(\frac{\mu_i^t(+1)}{\mu_i^t(-1)}\right) = z_i^{t-1} + x_{ij} \log\left(\frac{\bar{p}_j^t}{1-\bar{p}_j^t}\right) \qquad \text{where} \qquad z_i^0 = \log\left(\frac{q}{1-q}\right) \qquad (7)$$

This has both the advantage of converting the chain of products into a summation, and removing the need of normalising the factors $\mu_i^t$. Second, we can use the current log-odds $\boldsymbol{z}^t$ to compute the worker accuracy estimate as follows:

$$\bar{p}_j^t = \frac{\sum_{i \in M_j^{t-1}} \mathrm{sig}(x_{ij} z_i^{t-1}) + \alpha}{|M_j^{t-1}| + \alpha + \beta} \qquad \text{where} \qquad \mathrm{sig}(z_i^{t-1}) \equiv \frac{1}{1 + \exp(-z_i^{t-1})} = \mu_i^{t-1}(+1) \tag{8}$$

Thanks to the additive nature of Equation 7, we can quickly update the log-odds $\boldsymbol{z}^t$ as we observe new labels. More in detail, in Line 1 of Algorithm 1 we set $z_i^0$ to its prior value. Then, for every new label $x_{ij}$, we estimate the mean accuracy of worker $j$ given the current value of $\boldsymbol{z}^{t-1}$ (see Line 5), and add its contribution to the log-odds on task $i$ (see Line 6). In the end (Line 7), we compute the final predictions by selecting the maximum-a-posteriori class $\hat{y}_i^T = \mathrm{sign}(z_i^T)$.

This algorithm runs in $O(TL)$ time, where $L = \max_j(|M_j|)$ is the maximum number of labels per worker. This makes it particularly efficient in an online setting, e.g. under an adaptive collection strategy, since it takes only $O(L)$ operations to update its estimates after observing a new label. In Section 5 we show that its computational speed is on par with the simple majority voting scheme.

### 3.2 Sorted SBIC

In an offline setting, or when more computational resources are available, we have the opportunity of trading off some of the computational speed of Fast SBIC in exchange for better predictive accuracy. We can do so by running multiple copies of the algorithm in parallel, and presenting them the labels in $X$ in different orders. We show the implementation of this idea in Algorithm 2, which we call Sorted SBIC.

The intuition behind the algorithm is the following. When running Fast SBIC, the estimates $\hat{\boldsymbol{\mu}}^t$ and $\hat{\boldsymbol{\nu}}^t$ are very close to their prior in the first rounds. As time passes, two things change. First, we have more information since we observe more data points. Second, we run more updates on each factor $\mu_i^t$ and

---

**Algorithm 2** Sorted SBIC

---

**Input**: dataset $X$, availability $a$, policy $\pi$, prior $\theta$
**Output**: final predictions $\hat{y}^T$

1:   $s_i^k = \log(q/(1-q))$, $\forall i \in M, \forall k \in M$        8:    $z_i^t = \log(q/(1-q))$, $\forall i \in M$
2:   **for** $t = 1$ **to** $T$ **do**                          9:      **for** $u = 1$ **to** $t$ **do**
3:      $i \leftarrow \pi(t)$                              10:        $i \leftarrow \pi(u)$
4:      $j \leftarrow a(t)$                              11:        $j \leftarrow a(u)$
5:      **for all** $k \in M : k \neq i$ **do**
6:          $\bar{p}_j^k \leftarrow \frac{\sum_{h \in M_j^t \setminus k} \text{sig}(x_{hj} s_h^k) + \alpha}{|M_j^t \setminus k| + \alpha + \beta}$      12:        $\bar{p}_j^i \leftarrow \frac{\sum_{h \in M_j^t \setminus i} \text{sig}(x_{hj} s_h^i) + \alpha}{|M_j^t \setminus i| + \alpha + \beta}$

7:          $s_i^k \leftarrow s_i^k + x_{ij} \log(\bar{p}_j^k/(1-\bar{p}_j^k))$     13:        $z_i^t \leftarrow z_i^t + x_{ij} \log(\bar{p}_j^i/(1-\bar{p}_j^i))$

                                                                   14: **return** $\hat{y}_i^T = \text{sign}(z_i^T)$, $\forall i \in M$

---

$\nu_j^t$. Because of these, the estimates $\hat{\boldsymbol{\mu}}^t$ and $\hat{\boldsymbol{\nu}}^t$ become closer and closer to their ground-truth values. As a result, we get more accurate predictions on a specific task $i$ when the corresponding subset of labels is processed towards the end of the collection process ($t \approx T$), rather than the beginning ($t \approx 0$).

We exploit this property in Sorted SBIC by keeping a separate *view* of the log-odds $\boldsymbol{s}^k$ for each task $k \in M$ (see Line 1). Then, every time we observe a new label $x_{ij}$ we update the views for all tasks $k$ *except* the one we observed the label on (see Lines 5-7). We skip it because we want to process the corresponding label $x_{ij}$ at the end. Note that in Line 6 we compute a *different* estimate $\bar{p}_j^k$ for each task $k \neq i$. This is because we are implicitly running $|M|$ copies of Fast SBIC, and each copy can only see their correponding information stored in $\boldsymbol{s}^k$.

Finally, we need to process all the labels we skipped. If we are running Sorted SBIC offline, we only need to do so once at the end of the collection process. Conversely, in an online setting we need to repeat the same procedure at each time step $t$. Lines 8-13 contain the corresponding pseudocode. Notice how we compute the estimates $\bar{p}_j^i$ by looking at all the tasks $M_j^t$ labelled by worker $j$ *except* for task $i$ itself. This is because we skipped the corresponding label $x_{ij}$ in the past, and we are processing it right now.

The implementation of Sorted SBIC presented in Algorithm 2 runs in $O(|M|TL)$ time, which is a factor $|M|$ slower than Fast SBIC since we are running $|M|$ copies of it in parallel. By sharing the views $\boldsymbol{s}^k$ across different tasks, we can reduce the complexity to $O(\log(|M|)TL)$. However, this is only possible if the algorithm is run in an offline setting, where the whole dataset $X$ is known in advance. This additional time complexity comes with improved predictive accuracy. In Sections 4 and 5 we quantify such improvement both theoretically and empirically.

## 4 Theoretical analysis

In this section we study the predictive performance of SBIC from the theoretical perspective. As is the norm in the crowdsourcing literature, we establish an exponential relationship between the probability of an error and the average number of labels per task $R = T/|M|$ in the form $\mathbb{P}(\hat{y}_i \neq y_i) \leq \exp(-cR + o(1))$. Computing the constant $c$ is not trivial as its value depends not only on the properties of the crowd and the aggregation algorithm, but also the collection policy $\pi$ we use (see Section 2.2). In this regard, previous results are either very conservative [Karger *et al.*, 2014; Manino *et al.*, 2018], or assume a large number of labels per worker so that the estimates of $\boldsymbol{p}$ are close to their ground-truth value [Gao *et al.*, 2016].

Here, we take a different approach and provide exponential bounds that are both close to the empirical performance of SBIC, and valid for any number of labels per worker. We achieve this by focusing on the asymptotic case, where we assume that the predictions of SBIC are converging to the ground-truth after observing a large enough number of labels. More formally:

**Definition 1.** *For any small $\epsilon > 0$, define $t'$ as the minimum size of the dataset $X$, such that $\mu_i^{t'}(y_i) \geq 1 - \epsilon$ for any task $i \in M$ with high probability.*

For any larger dataset, when $t \geq t'$ the term $\mu_i^t(x_{ij})$ is very close to the indicator $\mathbb{I}(x_{ij} = y_i)$. As a consequence, we can replace the worker accuracy estimates in Equation 6 with $\bar{p}_j^t = (k_j^t + $

$\alpha)/(|M_j^t| + \alpha + \beta)$, where $k_j^t$ is the number of correct answers. With this in mind, we can establish the following bound on the performance of SBIC under the UNI policy:

**Theorem 1.** *For a crowd of workers with accuracy $p_j \sim Beta(\alpha, \beta)$, $L$ labels per worker, $R$ labels per task, the probability of an error under the UNI policy is bounded by:*

$$\mathbb{P}(\hat{y}_i \neq y_i) \leq \exp\big(-R\log F(L, \alpha, \beta) + o(1)\big), \qquad \text{for all } i \in M \tag{9}$$

*where $F(L, \alpha, \beta)$ depends on the variant of SBIC we use. For Sorted SBIC we have:*

$$F_{sorted}(L, \alpha, \beta) = \sum_{k=0}^{\bar{L}} \mathbb{P}(k|\bar{L}, \alpha, \beta) 2\sqrt{\left(\frac{k+\alpha}{\bar{L}+\alpha+\beta}\right)\left(\frac{\bar{L}-k+\beta}{\bar{L}+\alpha+\beta}\right)} \tag{10}$$

*where $\bar{L} = L - 1$, and the probability of observing $k$ is:*

$$\mathbb{P}(k|\bar{L}, \alpha, \beta) = \binom{\bar{L}}{k}\frac{B(k+\alpha, \bar{L}-k+\beta)}{B(\alpha, \beta)} \tag{11}$$

*For Fast SBIC we have instead:*

$$F_{fast}(L, \alpha, \beta) = \frac{1}{L}\sum_{h=1}^{L} F_{sorted}(h, \alpha, \beta) \tag{12}$$

For reasons of space, we only present the intuition behind Theorem 1 here (the full proof is in Appendix A). First, $\mathbb{P}(k|\bar{L}, \alpha, \beta)$ is the probability of observing a worker with accuracy $p_j \sim Beta(\alpha, \beta)$ produce $k$ correct labels over a total of $\bar{L}$ labels. Second, the square root term converges to the corresponding term $2\sqrt{p_j(1-p_j)}$ in [Gao *et al.*, 2016] when the estimates $\bar{p}_j^t$ become close to their ground-truth value $p_j$. Finally, the constant $F_{fast}$ is averaged over $\bar{L} \in [0, L-1]$ as this is the number of past labels we use to form each worker's estimate $\bar{p}_j^t$ during the execution of Fast SBIC.

Similarly, for the US policy we have the following theorem:

**Theorem 2.** *For a crowd of workers with accuracy $p_j \sim Beta(\alpha, \beta)$, $L$ labels per worker, an average of $R$ labels per task, and $|M| \to \infty$, the probability of an error under the US policy is bounded by:*

$$\mathbb{P}(\hat{y}_i \neq y_i) \leq \exp\big(-RG(L, \alpha, \beta) + o(1)\big), \qquad \text{for all } i \in M \tag{13}$$

*where $G(L, \alpha, \beta)$ depends on the variant of SBIC we use. For Sorted SBIC we have:*

$$G_{sorted}(L, \alpha, \beta) = \sum_{k=0}^{\bar{L}} \mathbb{P}(k|\bar{L}, \alpha, \beta) \log\left(\frac{k+\alpha}{\bar{L}-k+\beta}\right)\frac{(k+\alpha)-(\bar{L}-k+\beta)}{\bar{L}+\alpha+\beta} \tag{14}$$

*For Fast SBIC we have instead:*

$$G_{fast}(L, \alpha, \beta) = \frac{1}{L}\sum_{h=1}^{L} G_{sorted}(h, \alpha, \beta) \tag{15}$$

A full proof of Theorem 2 is in Appendix A. Here, note that the logarithm term corresponds to the log-odds of a worker with accuracy $\bar{p}_j^t$, and the right-most term is the expected value of a new label $x_{ij}$ provided by said worker.

In practice, both variants of SBIC reach the asymptotic regime described in Definition 1 for fairly small values of $R$. As an example, in Figure 2 we compare our theoretical results with the empirical performance of SBIC on synthetic data. There, we can see how the slope we predict in Theorems 1 and 2 closely matches the empirical decay in prediction error of SBIC. This in contrast with the corresponding state-of-the-art results in [Manino *et al.*, 2018], which apply to any state-of-the-art probabilistic inference algorithm (i.e. not MAJ) but are significantly more conservative.

# 5   Empirical analysis

In this section we compare the empirical performance of SBIC with the state-of-the-art algorithms listed in Section 2.3. Our analysis includes synthetic data, real-world data and a discussion on time complexity. For reasons of space, we report the details of the algorithm implementations and experiment parameters in Appendix B.

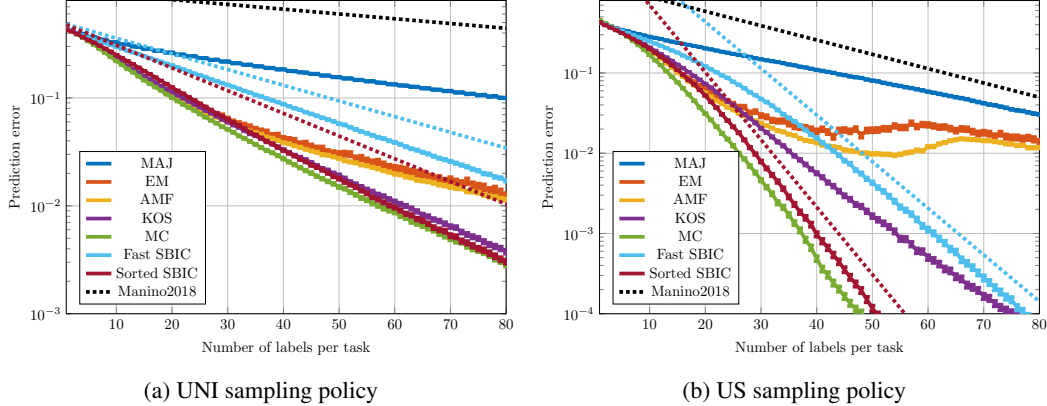

(a) UNI sampling policy  (b) US sampling policy

Figure 2: Prediction error on synthetic data with $p_j \sim \text{Beta}(\alpha, \beta)$, $q = 0.5$ and $L = 10$. The accuracy guarantees for SBIC are represented by a dotted line in the corresponding colour.

**Synthetic data.** First, we run the algorithms on synthetic data. With this choice we can make sure that the assumptions of the underlying one-coin Dawid-Skene model are met. In turn, this allows us to compare the empirical performance of SBIC with the theoretical results in Section 4.

To do so, we extract workers from a distribution $p_j \sim \text{Beta}(4, 3)$, representing a non-uniform population with large variance. Crucially, the mean of this distribution is above $\frac{1}{2}$, thus ensuring that the crowd is biased towards the correct answer. Additionally, we set the number of tasks to $M = 1000$ and the number of labels per worker to $L = 10$. This represents a medium-sized crowdsourcing project with a high worker turnout. Finally, we run EM, AMF, MC and SBIC with parameters $\alpha$ and $\beta$ matching the distribution of $p_j$. Conversely, MAJ and KOS do not require any extra parameter. We omit the results for TE since in this setting the task-worker matrix $X$ is too sparse for the algorithm to produce non-random predictions.

In Figures 2a and 2b we show the results obtained under the UNI and US sampling policies respectively. For reference, we also plot the bounds of Theorems 1 and 2 up to an arbitrary $o(1)$ constant (see Section 4 for the related discussion). As expected, the performance of all algorithms under the US policy greatly improves with respect to the UNI policy. Also, notice how MAJ is consistently outperformed by the other algorithms in this setting (this is not the case on real-world data, as we show below). Additionally, both variants of SBIC perform well, with Sorted SBIC achieving state-of-the-art performance under the UNI policy and matching the computationally-expensive MC algorithm under the US policy. Interestingly, Fast SBIC is asymptotically competitive as well, but suffers from an almost constant performance gap (in logarithmic scale). Finally, both EM and AMF tend to lose their competitiveness as the number of labels per task $R$ increases. This is due to their inability to form unbiased estimates of the workers' accuracy with few labels per worker. Under the US policy this may lead to poor sampling behaviour, which explains the lack of improvement in predictive accuracy for $R > 40$ in Figure 2b.

**Time complexity.** As we show in our experiments on synthetic data, all algorithms benefit from an adaptive sampling strategy. However, in order to deploy such policy we need to be able to update our estimates in real time, and only the MAJ and Fast SBIC algorithms are capable of that. To prove this point, we measure the average time the algorithms take to complete the simulations presented in Figure 2b, i.e. when used in conjunction with the US policy. We plot the results in Figure 3. Note how Fast SBIC matches MAJ in terms of computational speed, whereas all the other algorithms are orders of magnitude slower. This makes Fast SBIC the only viable alternative to MAJ for the online setting, particularly because it can deliver superior predictive accuracy.

**Real-world data.** Second, we consider the 5 publicly available dataset listed in Table 1, which come with binary annotations and ground-truth values. For more information on the datasets see [Snow *et al.*, 2008; Welinder *et al.*, 2010; Lease and Kazai, 2011]. The performance of the algorithms is reported in Table 2. There we run EM, AFM, MC and SBIC with the generic prior $\alpha = 2$, $\beta = 1$ and $q = \frac{1}{2}$ as proposed in Liu *et al.* [2012]. Additionally, we include the triangular estimation (TE)

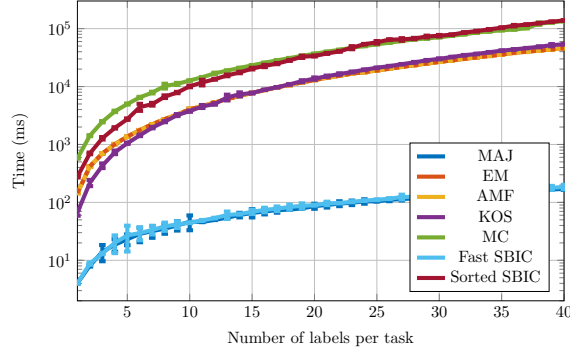

Figure 3: Time required to complete a single run with $|M| = 1000$ tasks under the US policy.

algorithm from Bonald and Combes [2017], since it outputs non-random predictions on most of the aforementioned datasets.

Table 1: Summary of the real-world datasets

| Dataset | # Tasks | # Workers | # Labels | Avg. $L$ | Avg. $R$ |
|---------|---------|-----------|----------|----------|----------|
| Birds | 108 | 39 | 4212 | 108 | 39 |
| Ducks | 240 | 53 | 9600 | 181 | 40 |
| RTE | 800 | 164 | 8000 | 49 | 10 |
| TEMP | 462 | 76 | 4620 | 61 | 10 |
| TREC | 711 | 181 | 2199 | 12 | 3 |

Table 2: Prediction error on the real-world datasets

| Dataset | MAJ | EM | AMF | KOS | MC | Fast SBIC | Sorted SBIC | TE |
|---------|-----|-----|-----|-----|-----|-----------|-------------|-----|
| Birds | 0.241 | 0.278 | 0.278 | 0.278 | 0.341 | 0.260 | 0.298 | **0.194** |
| Ducks | **0.306** | 0.412 | 0.412 | 0.396 | 0.412 | 0.400 | 0.405 | 0.408 |
| RTE | 0.100 | **0.072** | 0.075 | 0.491 | 0.079 | 0.075 | **0.072** | 0.257 |
| TEMP | **0.057** | 0.061 | 0.061 | 0.567 | 0.095 | 0.059 | 0.062 | 0.115 |
| TREC | 0.257 | **0.217** | 0.266 | 0.259 | 0.302 | 0.251 | 0.239 | 0.451 |

Interestingly, the MAJ algorithm performs quite well and achieves the best score on the Ducks and TEMP datasets. This confirms the practitioner's knowledge that majority voting is a robust and viable algorithms in most settings. Unsurprisingly, TE achieves its best score on the Birds dataset, which has a full task-worker matrix $X$. On the contrary, its predictions are almost random on the TREC dataset, which has a low number of labels per worker. Finally, both variants of SBIC match the performance of the other state-of-the-art Bayesian algorithms (EM, AFM, MC), with Sorted SBIC achieving the best score on RTE, and EM on both RTE and TREC. More importantly, Fast SBIC is always close to the other algorithms, making a strong case for its computationally efficient approach to Variational Bayes.

## 6 Conclusions

In this paper we proposed Streaming Bayesian Inference for Crowdsourcing, a new method to infer the ground-truth from binary crowdsourced data. This method combines strong theoretical guarantees, state-of-the-art accuracy and computational efficiency. The latter makes it the only viable alternative to majority voting when real-time decisions need to be made in an online setting. We plan to extend these techniques to the multi-class case as our future work.

**Acknowledgments**

This research is funded by the UK Research Council project ORCHID, grant EP/I011587/1. The authors acknowledge the use of the IRIDIS High Performance Computing Facility, and associated support services at the University of Southampton.

## Footnotes

[1] www.mturk.com

[2] www.figure-eight.com

[3]Exact knowledge of $\alpha$, $\beta$ and $q$ is not necessary in practice. See Section 5 for examples.

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
