[Supplementary Material]

# Streaming Bayesian Inference for Crowdsourced Classification (Appendices)

**Edoardo Manino**
University of Southampton
E.Manino@soton.ac.uk

**Long Tran-Thanh**
University of Southampton
l.tran-thanh@soton.ac.uk

**Nicholas R. Jennings**
Imperial College, London
n.jennings@imperial.ac.uk

## Appendix A - proofs

This appendix contains the proofs of Theorems 1 and 2 (see Section 4 in the main paper).

**Proof of Theorem 1.**

In the following discussion, we omit the index $t$ for simplicity. By definition, the probability of an error on task $i$ is:

$$\mathbb{P}(\hat{y}_i \neq y_i | q) = q\mathbb{P}(\hat{y}_i = -1 | y_i = +1) + (1-q)\mathbb{P}(\hat{y}_i = +1 | y_i = -1) \tag{16}$$

Now, assume that we know both the workers' accuracy $\boldsymbol{p}$ and the estimates $\bar{\boldsymbol{p}}$. Also, define the halved log-odds on task $i$ as $h_i \equiv \frac{1}{2}w_q + \sum_{j \in N_i} x_{ij}\frac{1}{2}w_j$, where $w_q = \log(q/(1-q))$ and $w_j = \log(\bar{p}_j/(1-\bar{p}_j))$. Then, the conditional probability of a classification error is the following:

$$\mathbb{P}(\hat{y}_i = -1 | y_i = +1, \boldsymbol{p}, \bar{\boldsymbol{p}}) = \sum_{X_i} \left( \mathbb{I}\{h_i < 0\} + \frac{1}{2}\mathbb{I}\{h_i = 0\} \right) \mathbb{P}(X_i | y_i = +1, \boldsymbol{p}) \tag{17}$$

$$\mathbb{P}(\hat{y}_i = +1 | y_i = -1, \boldsymbol{p}, \bar{\boldsymbol{p}}) = \sum_{X_i} \left( \mathbb{I}\{h_i > 0\} + \frac{1}{2}\mathbb{I}\{h_i = 0\} \right) \mathbb{P}(X_i | y_i = -1, \boldsymbol{p}) \tag{18}$$

where $X_i$ is the subset of labels cast on task $i$. Let us write the conditional probability of observing $X_i$ explicitly:

$$\begin{aligned}
\mathbb{P}(X_i | y_i = +1, \boldsymbol{p}) &= \prod_{j \in N_i} \mathbb{P}(x_{ij} | y_i = +1, p_j) \\
&= \prod_{j \in N_i} \frac{\mathbb{P}(x_{ij} | y_i = +1, p_j) f(x_{ij}) \exp(-x_{ij}\frac{1}{2}w_j) \sqrt{\bar{p}_j(1-\bar{p}_j)}}{f(x_{ij}) \sqrt{\bar{p}_j(1-\bar{p}_j)} \exp(-x_{ij}\frac{1}{2}w_j)} \\
&= \exp(h_i - \frac{1}{2}w_q) \prod_{j \in N_i} \frac{\mathbb{P}(x_{ij} | y_i = +1, p_j)}{f(x_{ij})} \sqrt{\bar{p}_j(1-\bar{p}_j)}
\end{aligned} \tag{19}$$

where $f(+1) \equiv \bar{p}_j$ and $f(-1) \equiv 1 - \bar{p}_j$. Similarly:

$$\mathbb{P}(X_i | y_i = -1, \boldsymbol{p}) = \exp(-h_i + \frac{1}{2}w_q) \prod_{j \in N_i} \frac{\mathbb{P}(x_{ij} | y_i = -1, p_j)}{f(-x_{ij})} \sqrt{\bar{p}_j(1-\bar{p}_j)} \tag{20}$$

By substituting Equation 19 in Equation 17 we get:

$$\mathbb{P}(\hat{y}_i = -1 | y_i = +1, \boldsymbol{p}, \bar{\boldsymbol{p}}) \leq \exp(-\frac{1}{2}w_q) \sum_{X_i} \prod_{j \in N_i} \frac{\mathbb{P}(x_{ij} | y_i = +1, p_j)}{f(x_{ij})} \sqrt{\bar{p}_j(1 - \bar{p}_j)}$$

$$= \exp(-\frac{1}{2}w_q) \prod_{j \in N_i} \left(\frac{p_j}{\bar{p}_j} + \frac{1 - p_j}{1 - \bar{p}_j}\right) \sqrt{\bar{p}_j(1 - \bar{p}_j)} \quad (21)$$

since $\exp(h_i) \leq 1$ for all $h_i \leq 0$. Similarly, substituting Equation 20 in Equation 18 yields:

$$\mathbb{P}(\hat{y}_i = +1 | y_i = -1, \boldsymbol{p}, \bar{\boldsymbol{p}}) \leq \exp(\frac{1}{2}w_q) \prod_{j \in N_i} \left(\frac{p_j}{\bar{p}_j} + \frac{1 - p_j}{1 - \bar{p}_j}\right) \sqrt{\bar{p}_j(1 - \bar{p}_j)} \quad (22)$$

By combining, Equations 21 and 22 according to Equation 16 we get the following:

$$\mathbb{P}(\hat{y}_i \neq y_i | q, \boldsymbol{p}, \bar{\boldsymbol{p}}) \leq 2\sqrt{q(1 - q)} \prod_{j \in N_i} \left(\frac{p_j}{\bar{p}_j} + \frac{1 - p_j}{1 - \bar{p}_j}\right) \sqrt{\bar{p}_j(1 - \bar{p}_j)} \quad (23)$$

which is valid for any prior on the task class $q \in (0, 1)$, any worker accuracy $p_j \in [0, 1]$, and any estimate $\bar{p}_j \in (0, 1)$.

Note that under the assumptions in Definition 1 (see main paper), the accuracy estimate $\bar{p}_j$ depends on the number of worker's $j$ correct answers. With this knowledge, we can compute the expected zero-one loss across all instances of the crowd $\boldsymbol{p}$ and the estimates $\bar{\boldsymbol{p}}$. Specifically, the expectation of Equation 23 yields the following:

$$\mathbb{P}(\hat{y}_i \neq y_i | q) = \mathbb{E}_{\boldsymbol{p}, \bar{\boldsymbol{p}}}\{\mathbb{P}(\hat{y}_i \neq y_i | q, \boldsymbol{p}, \bar{\boldsymbol{p}})\}$$

$$= \mathbb{E}_{\boldsymbol{p}, X}\{\mathbb{P}(\hat{y}_i \neq y_i | q, \boldsymbol{p}, \bar{\boldsymbol{p}})\}$$

$$\leq 2\sqrt{q(1 - q)} \prod_{j \in N_i} \mathbb{E}_{X_j}\left\{\left(\frac{\mathbb{E}_{p_j | X_j}\{p_j\}}{\bar{p}_j} + \frac{1 - \mathbb{E}_{p_j | X_j}\{p_j\}}{1 - \bar{p}_j}\right) \sqrt{\bar{p}_j(1 - \bar{p}_j)}\right\} \quad (24)$$

$$= 2\sqrt{q(1 - q)} \prod_{j \in N_i} \mathbb{E}_{X_j}\left\{2\sqrt{\bar{p}_j(1 - \bar{p}_j)}\right\}$$

where $X_j$ is the subset of labels provided by worker $j$ except for $x_{ij}$, and $\mathbb{E}_{p_j | X_j}\{p_j\} = \bar{p}_j$ by definition because $\bar{p}_j$ is the exact mean of the posterior of a beta-distributed Bernoulli variable with $|X_j|$ observations.

Finally, we can compute the value of the remaining expectation over $X_j$ by considering how the output of each worker is used in our algorithms. In Sorted SBIC, each worker provides $\bar{L} = L - 1$ labels on tasks other than $i$, before casting their final vote on task $i$. As a consequence we have:

$$F_{sorted}(L, \alpha, \beta) \equiv \mathbb{E}_{X_j}\left\{2\sqrt{\bar{p}_j(1 - \bar{p}_j)}\right\}$$

$$= 2\sum_{k=0}^{\bar{L}} \mathbb{P}\left(\sum_{i' \in M_j \setminus i} \mathbb{I}\{x_{i'j} = y_{i'}\} = k\right) \sqrt{\left(\frac{k + \alpha}{\bar{L} + \alpha + \beta}\right)\left(\frac{\bar{L} - k + \beta}{\bar{L} + \alpha + \beta}\right)} \quad (25)$$

where the probability of observing $k$ correct answers out of $\bar{L}$ can be computed according to the prior $p_j \sim \text{Beta}(\alpha, \beta)$, and leads to the result in the theorem.

In contrast, the Fast SBIC algorithm computes the estimates $\bar{p}_j$ on a number of labels across the range $[0, L - 1]$ in equal proportions. We can compute the expectation over $X_j$ for each of these values separately according to Equation 25, and then take the average. The result of this operation yields the formula for $F_{fast}(L, \alpha, \beta)$ shown in the theorem. $\qquad \square$

**Proof of Theorem 2.**

From the perspective of a single task $i$, the US policy operates in short bursts of activity, as $i$ keeps receiving new labels until it is no longer the most uncertain one. We define $z_T \equiv \max_t\{\min_i\{|z_i|\}\}$ as the largest threshold that all tasks have crossed at some point of the collection process. In this respect, we can model the evolution of the log-odds $z_i^t$ as a bounded random walk, which starts

from the prior value $z_i^0 = w_q$ where $w_q = \log(q/(1-q))$, and ends when $z_i^t$ leaves the interval $(-z_T, +z_T)$.

Given this, let us assume that we can fix the threshold $z_T > |w_q|$ and then collect as many labels as needed in order to cross it. We denote the log-odds after crossing the threshold as $z_i^r$, where $z_i^r \notin (-z_T, +z_T)$, and the log-odds at the step before as $z_i^{r-1}$. According to this definition, $r$ is a stopping time since it is uniquely defined by the information collected before step $r$. Thus, we can use Wald's equation to link the expected value of $z_i^r$ and the stopping time $r$:

$$\mathbb{E}\{z_i^r\} = \mathbb{E}\{r\}\mathbb{E}\{x_{ij}w_j\} + w_q \tag{26}$$

where $w_j = \log(\bar{p}_j/(1-\bar{p}_j))$ is the weight associated to each worker.

Recall, however, that $z_i^r$ is the sum of $r$ i.i.d random variables, and that $z_i^{r-1} \in (-z_T, +z_T)$ by definition. As a consequence, we can further bound the expected value of $z_i^r$ (conditioned on the ground-truth $y_i$) as follows:

$$\mathbb{E}\{z_i^r|y_i = +1\} = \mathbb{E}\{z_i^{r-1}|y_i = +1\} + \mathbb{E}\{x_{ij}w_j|y_i = +1\} < +z_T + \mathbb{E}\{x_{ij}w_j|y_i = +1\} \tag{27}$$

$$\mathbb{E}\{z_i^r|y_i = -1\} = \mathbb{E}\{z_i^{r-1}|y_i = -1\} + \mathbb{E}\{x_{ij}w_j|y_i = -1\} > -z_T + \mathbb{E}\{x_{ij}w_j|y_i = -1\} \tag{28}$$

By plugging Equations 27 and 28 into Equation 26, we can derive the following bounds on the expected number of steps $r$ we need to reach the threshold $z_T$:

$$\mathbb{E}\{r|y_i = +1\} < \frac{z_T + \mathbb{E}\{x_{ij}w_j|y_i = +1\} - w_q}{\mathbb{E}\{x_{ij}w_j|y_i = +1\}} \tag{29}$$

$$\mathbb{E}\{r|y_i = -1\} < \frac{z_T + \mathbb{E}\{x_{ij}w_j|y_i = +1\} - w_q}{\mathbb{E}\{x_{ij}w_j|y_i = +1\}} \tag{30}$$

where we used the fact that $\mathbb{E}\{x_{ij}w_j|y_i = -1\} = -\mathbb{E}\{x_{ij}w_j|y_i = +1\}$. And finally:

$$\mathbb{E}\{r\} = q\mathbb{E}\{r|y_i = +1\} + (1-q)\mathbb{E}\{r|y_i = -1\} < \frac{z_T + \mathbb{E}\{x_{ij}w_j|y_i = +1\} + (1-2q)w_q}{\mathbb{E}\{x_{ij}w_j|y_i = +1\}} \tag{31}$$

At the same time, we also know that the random walks on the $|M|$ tasks are independent, and that the variance of $r$ for a bounded random walk with i.i.d. steps is finite. Therefore, as $|M| \to \infty$ the total number of steps required to cross the threshold on all the tasks will converge to its expected value, i.e. $T \to |M|\mathbb{E}\{r\}$. This property allows us to substitute $\mathbb{E}\{r\} = T/|M| = R$ and get a bound on the value of the threshold $z_T$ given the average number of labels per task $R$:

$$z_T > (2q-1)w_q + (R-1)\mathbb{E}\{x_{ij}w_j|y_i = +1\} \tag{32}$$

Having a value for the threshold $z_T$ is crucial because it relates to the probability of a classification error. In fact, under the assumption that $p_j \sim \text{Beta}(\alpha, \beta)$ and $\mu_i(y_i) \to 1$, our estimates of the workers' accuracy satisfy the condition $\mathbb{E}_{p_j|X_j}\{p_j\} = \bar{p}_j$. Thus, we can establish the following equality:

$$\mathbb{P}(\hat{y}_i \neq y_i|q) = \mathbb{E}\{\text{sig}(-|z_i|)\} \tag{33}$$

and now, since we know that $|z_i| > z_T$, we can use Equation 32 to bound the probability of an error as follows:

$$\mathbb{P}(\hat{y}_i \neq y_i|q) < \text{sig}(-z_T)$$
$$\leq \exp\left(-(2q-1)w_q - (R-1)\mathbb{E}\{x_{ij}w_j|y_i = +1\}\right) \tag{34}$$

Finally, we can compute the expected value of $x_{ij}w_j$ over the true and estimated accuracy $p_j, \bar{p}_j$ of each worker. The results depends on how the individual variant of SBIC computes the estimates $\bar{p}$. In Sorted SBIC, each worker provides $\bar{L} = L - 1$ labels on tasks other than $i$, before casting their final vote on task $i$. As a consequence we have:

$$G_{sorted}(L, \alpha, \beta) \equiv \mathbb{E}_{p_j, \bar{p}_j}\{x_{ij}w_j|y_i = +1\}$$

$$= \sum_{k=0}^{\bar{L}} \mathbb{P}\left(\sum_{i' \in M_j \setminus i} \mathbb{I}\{x_{i'j} = y_{i'}\} = k\right) \log\left(\frac{k+\alpha}{\bar{L}-k+\beta}\right) \frac{(k+\alpha) - (\bar{L}-k+\beta)}{\bar{L}+\alpha+\beta} \tag{35}$$

where the last term takes into account the value of $x_{ij}$, the logarithm the value of $w_j$, and the probability of observing $k$ correct answers out of $\bar{L}$ can be computed according to the prior $p_j \sim \text{Beta}(\alpha, \beta)$, and leads to the result in the theorem.

As in the proof of Theorem 1, the value $G_{fast}(L, \alpha, \beta)$ for the Fast SBIC algorithm can be derived from Equation 35 as shown in the statement of the present theorem. $\square$

## Appendix B - experimental setup

In this appendix we list the implementation choices and parameter values we used in our experimental setup. We begin with a description of the data we use in Section 5.

**Synthetic data.** Given the values of $|M|$, $L$ and $R$, we generate a crowd of $|N| = |M|R/L$ workers by extracting them from the distribution $p_j \sim \text{Beta}(\alpha, \beta)$. Then, for each worker $j$ we extract $L$ answers according to their true accuracy $p_j$ and the ground-truth $\boldsymbol{y}$ which we set by convention to $y_i = +1, \forall i$. The assignment of the labels to the task is chosen at runtime according to the policy $\pi$ selected for the experiment. For each tuple $(R, L, \pi, algorithm)$ we run multiple experiments until we have 1000 runs that produced at least one classification error and we average the result. In this way, we can have low-variance estimates of the probability of an error even for large $R$. The error bars reported in Figure 1 (main paper) are computed with the Agresti-Coull method [Brown *et al.*, 2001] set at 99% confidence, and their value is as small as $10^{-5}$ (which makes them barely visible in our plots).

**Time complexity.** This set of experiments uses the same parameters of the synthetic data ones under the US policy. The only exception is that we average the execution time of the algorithms over 10 runs, and we report the empirical mean and standard deviation. The EM and AMF algorithms share the same implementation (albeit different parameters, as we explain below), and thus take the same time to execute.

**Real-world data.** We run the algorithms on the full datasets. Since Fast SBIC and Sorted SBIC are affected by the order in which they process the data, we shuffle the datasets and repeat the inference 100 times. Similarly, since MC is a stochastic algorithm, we repeat the sampling 100 times with different seeds. The results reported in Table 2 of the main paper are the average of these runs.

Next, we list the details of the inference algorithms.

**MAJ.** We use a straightforward implementation of majority voting. Under the US policy, we use the partial sum of votes $\sum_{j \in N_i} x_{ij}$ as an indication of uncertainty.

**AMF.** For experiments on fully-observed data or in conjunction with the UNI policy, we initialise the worker estimates to their mean prior value $\bar{p} = \alpha/(\alpha + \beta)$ and run 50 iterations of the algorithm to ensure convergence. For adaptive settings in conjunction with the US policy, we run 4 iterations after collecting each new label $x_{ij}$ to update the current estimates. At the end of the collection process we run 50 iterations from scratch. As for MC and SBIC, we use a matching prior $\alpha = 4, \beta = 3$ for synthetic data, a generic prior $\alpha = 2, \beta = 1$ for real-world data, and $q = \frac{1}{2}$ for all experiments.

**EM.** This algorithm shares the same implementation of AMF. The only difference is that we use $\bar{\alpha} = \alpha - 1$ and $\bar{\beta} = \beta - 1$ for the workers' prior. As explained in [Liu *et al.*, 2012], this forces the algorithm to compute the mode rather than the mean of the posterior worker accuracy.

**KOS.** We implement the algorithm as a power law iteration with alternating steps $\boldsymbol{w} = X\boldsymbol{z}$ and $\boldsymbol{z} = X\boldsymbol{w}$, where $\boldsymbol{w} = (w_1, \ldots, w_{|N|})$ are the worker weights and $\boldsymbol{z}$ are the task log-odds. This is the setup recommended by Karger *et al.* [2014] to achieve maximum performance, as opposed to the more theoretical-sound belief propagation algorithm. We initialise $\boldsymbol{z}$ to its majority voting value, and normalise the result at every iteration to prevent numerical explosion. For synthetic data (both under the UNI and US policies), we run only 5 power law iterations before producing the final estimates, as we found this yields better accuracy. For real-data we let the algorithm run for 100 iterations to reach convergence instead.

**MC.** We use two different implementation of this algorithm. For experiments on fully-observed data or in conjunction with the UNI policy, we use Gibbs Sampling [Murphy, 2012]. We initialise the chain according to the prior, and then update the variables $p, y$ for 500 steps and take the average across all of the samples. For experiments under the US policy, this setup is too slow. Thus, we implement a particle filter [Chopin, 2002] with 50 particles extracted according to the prior on $y$. We marginalise over the workers' accuracy $p$ to reduce the state space. After each new label $x_{ij}$ is received, we update the weights of the particles according to importance sampling. Every 10 labels we perform a full Gibbs step over all the particles and reinitialise the weights to 1. Across all experiments on synthetic data, we use a matching prior with $\alpha = 4$ and $\beta = 3$. For real-world data the prior is unknown, thus we use a generic setup with $\alpha = 2$ and $\beta = 1$ as suggested for Bayesian methods in [Liu *et al.*, 2012]. Finally, we use $q = \frac{1}{2}$ for all experiments.

**SBIC.** We use a straightforward implementation of the algorithms presented in Sections 3.1 and 3.2 of the main paper. As for MC and AMF, we use a matching prior $\alpha = 4, \beta = 3$ for synthetic data, a generic prior $\alpha = 2, \beta = 1$ for real-world data, and $q = \frac{1}{2}$ for all experiments.

**TE.** We use a straightforward implementation of the algorithm in [Bonald and Combes, 2017]. No parameters are required to run this algorithm.