[Reviews · NeurIPS 2019]

Reviewer 1



This is an interesting paper, and well written. Overall I like the contributions. I have the following comments to consider. I am not sure "feedforward" is an appropriate prefix for the technique, as it seems to suggest that the approach is feedforward neural networks based. Though, it is completely upto the authors. If I understand correctly, the proposed approach differs from the prior well known technique, one-coin Dawid-Skene model, in the sense that there is a prior distribution on worker accuracies. I suppose, similar prior model has been considered in other works? In section 3, although references are provided for known concepts used in the proposed technique, it doesn't give a clear picture on the novelty of the technique. May be, there should be a related work section. It is claimed that the proposed technique differs from the previous works, in terms of employing variational mean field approach, in the sense that KL divergence is minimized between the true and approximate posterior. It would be good to provide expressions for KL divergence, and to show how it becomes tractable to minimize it for the assumed prior distributions (so the posterior) and the mean field approximation of the posterior. In the previous works, coordinate decent is used for optimization, whereas it is proposed to perform (stochastic) updates, using a simple observation in each step, leading to local analytical optimizations. I suppose, the updates become analytical because of the specified beta distribution for the prior? Is it necessary to perform updates using a single observation? That seems like an extreme scenario even for online settings. How about updating with small sized batches of observations? I would like to mention it here that, in the recent literature of machine learning, information theoretic objectives like KL-divergence, mutual information have been used for optimization even if not computable analytically. And, it is a standard practice to use stochastic gradient rather than gradient. While it is interesting to see similar concepts being used for Bayesian inference, it should not come as a big surprise in terms of novelty. A small suggestions: section 3.2 could benefit from a sketch so as to explain the reasoning behind reordering the labels. In Theorem 1 and 2, the expressions for F(.) are too complicated to have clear interpretations, though useful to compute the upper bounds numerically. The authors do not provide any statements or remarks regarding the interpretability of the theorems. May be, the expressions should be simplified in the original theorems or in the corollaries, just a thought. In section 5, for synthetic data, Monte Carlo sampling seem to be outperforming the others. Though, its performance is not as good for the real datasets. Any intuitions on why that is the case? Also, I didn't see a reference for MC method. Is it true that the value of parameters, alpha, beta, are assumed to be known, rather than optimized or tuned? There are no errors bars, or statistical significance numbers for the experimental results presented in Section 5. I don't know how impactful the empirical evaluation is. I don't see significant improvement in accuracy numbers. What does this statement mean: "average accuracy just above 0.5"? ----------------- The rebuttal clarifies on the questions asked in the review well. I like this paper, and vote for its acceptance. Also, increasing the score from 7 to 8.

Reviewer 2



Overview: This paper proposes a Bayesian algorithm to aggregate the labels in a crowdsourced application. The authors consider the standard Dawid-Skene one coin model, in which each worker j is associated with parameter p_j, which represents the probability of the worker correctly assigning a label. In the Bayesian setting, the authors assume that the parameters p_j’s follow a Beta distribution. Computing the exact posterior distribution is computationally difficult. Therefore, the authors use variational approximation in which the posterior distribution is assumed to have a product form. Even with the variational approximation, parameter inference is computationally difficult. Therefore, the authors propose to perform a single coordinate descent step as opposed to an expensive coordinate descent step. The authors derive bounds for error rates. They also demonstrate the efficacy of their methods on both synthetic and real-world datasets. Strengths of the paper: The paper is tackling the classic problem of aggregating labels in a crowdsourced application. The paper is focusing on speed. The algorithms proposed are fast and simple to implement. Plus, they come with theoretical guarantees on the bounds for error rates. The fact that the theoretical bounds are close to empirical error rates is impressive. Weaknesses: The paper has the following main weaknesses: 1. The paper starts with the objective of designing fast label aggregation algorithms for a streaming setting. But it doesn’t spend any time motivating the applications in which such algorithms are needed. All the datasets used in the empirical analysis are static datasets. For the paper to be useful, the problem considered should be well motivated. 2. It appears that the output from the algorithm depends on the order in which the data are processed. This should be clarified. 3. The theoretical results are presented under the assumption that the predictions of FBI converge to the ground truth. Why should this assumption be true? It is not clear to me how this assumption is valid for finite R. This needs to clarified/justified. 3. The takeaways from the empirical analysis are not fully clear. It appears that the big advantage of the proposed methods is their speed. However, the experiments don’t seem to be explicitly making this point (the running times are reported in the appendix; perhaps they should be moved to the main body). Plus, the paper is lacking the key EM benchmark. Also, perhaps the authors should use a different dataset in which speed is most important to showcase the benefits of this approach. Update after the author response: I read the author rebuttal. I suggest the authors to add the clarifications they detailed in the rebuttal to the final paper. Update after the author response: I read the author rebuttal. I suggest the authors to add the clarifications they detailed in the rebuttal to the final paper. Also, the motivating crowdsourcing application where speed is really important is not completely clear to me from the rebuttal. I suggest the authors clarify this properly in the final paper.

Reviewer 3



The authors proposed Feedforward Bayesian Inference (FBI) for aggregating crowdsourced annotations in binary tasks. FBI follows the one-coin DS model, but adopts a streaming variational Bayes approach. It has two variants, namely Fast FBI and Sorted FBI. Fast FBI could run as fast as majority voting, while Sorted FBI archives better performance but runs slower. Bounds for the asymptotic accuracy of the two variants in both non-adaptive and adaptive settings are provided. Empirical results validates the bounds and show the efficiency and effectiveness of the two variants. The paper is well-organized and easy to follow. Efficient/real-time truth inference for crowdsourcing is an important and relevant problem. The proposed FBI algorithms are novel and are suitable for streaming settings but don’t achieve the best performance on average when all labels have been collected. Some questions: In practice, the prior distribution q may not be known in most cases. How will it affect the FBI algorithms if q is not known? How does Sorted FBI work in the streaming setting? From Algorithm 2, it seems that the Fast-FBI algorithm is only called when all labels have been collected (because the call is outside the t loop). What would Algorithm 2 look like if the t loop is the outer loop?

[Author Response · NeurIPS 2019]

**Reply to Reviewer 1.**  Most existing works put a prior on the worker accuracies too. Furthermore, variational mean field is used for the one-coin DS model by Liu *et al.* 2012. The novelty in our paper is how we optimise over the KL divergence in Eq. 5. In order to get an analytical form, we need both the Beta prior *and* a single-data update. Note that, while FBI can be *interpreted* as a form of stochastic GA, it does not correspond to its classic framework.

Other remarks: first, we will add a figure on the label reordering. Second, $F(\bullet)$ and $G(\bullet)$ have a probabilistic interpretation, but it only becomes clear after going through the proofs. Third, MC computes the true posterior. This is optimal when the model assumptions are met (synthetic data), but less robust when they are not (real data). References and implementation details are in Appendix C. Fourth, common practice in crowdsourcing is setting the uninformative prior $\alpha = 2, \beta = 1$. Fifth, the error bars are present but small (see Appendix C). Sixth, refer to our reply to reviewer 2 (point 4). Finally, we perform crowdsourcing when the workers, on average, are correct more often than not.

**Reply to Reviewer 2.**  1. A streaming algorithm allows the use of the US policy. This yields provably better accuracy than non-adaptive policies in crowdsourcing applications (Manino *et al.* 2018). We confirm this for FBI on synthetic data in Figure 1. To the best of our knowledge, there are no publicly available non-static datasets.

2. FBI relies on estimating the worker accuracies given the data observed so far (Eq. 7). If we change the order of the data, the sequence of estimates changes. This can affect the output of the algorithm. We will add a figure to illustrate this.

3. Our assumption holds when either the workers are very accurate or $R$ grows large. For example, in Figure 1b the theoretical error slope is attained for $R$ as small as 30. We will clarify this in the paper.

4. Our algorithm is designed for speed and theoretical guarantees. The empirical analysis shows that its accuracy is similar to the state of the art. We will move the running times to Section 5. Also, we will add the EM benchmark (see table below). On synthetic data, EM is slightly worse than AMF.

| Dataset | KOS | MAJ | MC | Sorted FBI | Fast FBI | AMF | TE | EM |
|---|---|---|---|---|---|---|---|---|
| Birds | 0.278 | 0.241 | 0.341 | 0.298 | 0.260 | 0.278 | **0.194** | 0.278 |
| Ducks | 0.396 | **0.306** | 0.412 | 0.405 | 0.400 | 0.412 | 0.408 | 0.412 |
| RTE | 0.491 | 0.100 | 0.079 | **0.072** | 0.075 | 0.075 | 0.257 | **0.072** |
| TEMP | 0.567 | **0.057** | 0.095 | 0.062 | 0.059 | 0.061 | 0.115 | 0.061 |
| TREC | 0.259 | 0.257 | 0.302 | 0.239 | 0.251 | 0.266 | 0.451 | **0.217** |

**Reply to Reviewer 3.**  Referring to Equation 5, the value of $q$ sets the starting value of the log-odds $z_i^0$. If we have prior information, we can use $q$ to bias $z_i^0$. Otherwise, we can set an uninformative prior $q = \frac{1}{2}$ and thus $z_i^0 = 0$. We use the latter in all our experiments.

To work in a streaming setting, Algorithm 2 needs to be implemented differently (see below). There we maintain a separate *view* of the log-odds $\boldsymbol{v}^k$ for each task $k \in M$, and use it to run $|M|$ copies of Fast FBI in parallel. Note that in lines 9-13 we are computing the *real* log-odds by processing all data points we skipped in lines 2-8. This operation has to be performed for every $t$. Since the online version takes more space to explain, we will dedicate an appendix to it.

---
**Algorithm 1** Sorted FBI (online version)

---
**Input**: dataset $X$, availability $a$, policy $\pi$, prior $\theta$
**Output**: predictions $\hat{\boldsymbol{y}}$

1: $v_i^k = \log(q/(1-q))$, $\forall i, k \in M$
2: **for** $t = 1$ **to** $T$ **do**
3: &emsp; $i \leftarrow \pi(t)$
4: &emsp; $j \leftarrow a(t)$
5: &emsp; **for all** $k \in M : k \neq i$ **do**
6: &emsp;&emsp; $\bar{p}_j^k \leftarrow \frac{\sum_{h \in M_j^t \setminus k} \mathrm{sig}(x_{hj} v_h^k) + \alpha}{|M_j^t \setminus k| + \alpha + \beta}$
7: &emsp;&emsp; $v_i^k \leftarrow v_i^k + x_{ij} \log(\bar{p}_j^k/(1 - \bar{p}_j^k))$

8: $z_i^t = \log(q/(1-q))$, $\forall i \in M$
9: **for** $u = 1$ **to** $t$ **do**
10: &emsp; $i \leftarrow \pi(u)$
11: &emsp; $j \leftarrow a(u)$
12: &emsp; $\bar{p}_j^i \leftarrow \frac{\sum_{h \in M_j^t \setminus i} \mathrm{sig}(x_{hj} v_h^i) + \alpha}{|M_j^t \setminus i| + \alpha + \beta}$
13: &emsp; $z_i^t \leftarrow z_i^t + x_{ij} \log(\bar{p}_j^i/(1 - \bar{p}_j^i))$
14: **return** $\hat{y}_i = \mathrm{sign}(z_i^T)$, $\forall i$

---



[Meta-Review · NeurIPS 2019]

This paper proposes two algorithms for recovering ground truth labels in crowd sourcing tasks for binary classisification. The problem is formulated as an online Bayesian version of the Dawid & Skene model (with beta priors) which is quite natural. The algorithms are based on variational approximations of the posterior (i.e. they try to find the best approximation that is product distribution). From this approach two algorithms are derived. One is fast (i.e. O(1) per label) and less accurate. The other one is more accurate and but slower (still polynomial time). Both algorithms are analyzed under two policies that choose the next instance/task to be labeled. (Round robin picks the instance with least labels. The least informative picks instance with greatest spread of the posterior.) The paper has theoretical analysis of the convergence of algorithms under both policies. It also contains results of experiments on the online problem in which the two algorithms are compared with other existing algorithms. The paper is clearly written. The results in the paper are a nice contribution to the literature of crowd sourcing.